# Did the New French Regulation of Zolpidem Decrease the Problematic Consumption of Zolpidem? A Field Study among Users

**DOI:** 10.3390/ijerph19158920

**Published:** 2022-07-22

**Authors:** Edouard-Jules Laforgue, Morgane Rousselet, Antoine Claudon, Aurélie Aquizerate, Pascale Jolliet, Marion Istvan, Caroline Victorri-Vigneau

**Affiliations:** 1CHU Nantes, Service de Pharmacologie Clinique—Centre d’Evaluation et d’Information sur la Pharmacovigilance-Addictovigilance, Nantes Université, F-44000 Nantes, France; morgane.rousselet@chu-nantes.fr (M.R.); antoine.claudon@chu-nantes.fr (A.C.); aurelie.aquizerate@chu-nantes.fr (A.A.); pascale.jolliet@univ-nantes.fr (P.J.); marion.istvan@chu-nantes.fr (M.I.); caroline.vigneau@chu-nantes.fr (C.V.-V.); 2Univ Tours, CHU Nantes, INSERM, MethodS in Patient-Centered Outcomes & HEalth ResEarch, Nantes Université, SPHERE, F-44000 Nantes, France; 3CHU Nantes, UIC Psychiatrie et Santé Mentale, Nantes Université, F-44000 Nantes, France

**Keywords:** zolpidem, drug regulation, patient-related outcome, substance use disorder, problematic consumption

## Abstract

Background: The French national drug regulatory authority stated, in 2017, that a secured prescription pad must be used for zolpidem prescriptions. This study aimed to evaluate the evolution of the problematic consumption of zolpidem at the individual level since the new regulation. Methods: Two nationwide populations of at-risk users of zolpidem were recruited: one in general practitioner (GP) offices and one in specialized care centers dedicated to drug dependence (SCDDs). Participants were asked about their zolpidem consumption before and after the regulation change. The primary outcome was the evolution of problematic zolpidem consumption, as defined by at least one of the following criteria: overconsumption, fraudulent ways of obtaining, effects sought other than hypnotic, and modes of administration other than oral. Results: A total of 243 participants were included: 125 from GP offices and 118 from SCDDs. In the GP population, the prevalence of patients who were identified as problematic consumers decreased from 24.8% to 20.8% (*p* = 0.593), whereas the prevalence decreased from 73.7% to 51.7% in the SCDD population (*p* < 0.001). The most prevalent criteria for problematic status were overconsumption and fraudulent ways. Conclusions: The new French regulation of zolpidem had different impacts among two different populations of at-risk zolpidem consumers.

## 1. Introduction

Zolpidem, a short-half-life non-benzodiazepine imidazopyridine marketed as a hypnotic for transient sleep disorders, is massively used and prescribed worldwide [1]. Its mechanism of action is a positive allosteric modulation of the GABA_A_ receptor, enhancing the receptor activity. Unlike benzodiazepines, zolpidem has a highly specific affinity for the alpha-1 subunit of this receptor providing the hypnotic effect [2]. Many reports of zolpidem abuse and dependence have been produced in recent decades [3,4,5,6,7,8]. Pre-clinical studies in rodents and baboons showed tolerance to the effect after repeated administration, and withdrawal manifestations suggested the potential for zolpidem dependence [4]. Many authors highlighted evidence for physical dependence and withdrawal symptoms after zolpidem chronic treatment discontinuation in humans. Concerning this potential for dependence, subjects with alcohol or drug abuse history are more at risk [4]. In France, the surveillance of pharmacodependence is performed by a network of 13 centers for the evaluation and information on drug dependence (Centres d’Evaluation et d’Information sur la Pharmacodépendance-Addictovigilance (CEIP-A) [9]. Two surveys, covering the 1993–2002 and 2003–2010 periods [4,5], highlighted the constant and significant signal of zolpidem abuse and dependence. Looking at the individual characteristics of problematic consumers of zolpidem, two distinct patterns of patients tended to be shown. The first pattern consists of patients with chronic sleep disorders who are seeking a sedative effect that vanishes with time, due to tolerance. The second includes younger patients with a more frequent addictive history who seek an amphetamine-like stimulant effect, sometimes with megadoses of zolpidem [4,5,10,11,12].

The summary of the product characteristics of zolpidem defines its proper use as a single daily intake at bedtime, with a maximum duration (4 weeks—tapering included) and daily dose (10 mg). It was updated after the first survey of the CEIP-A of Nantes in 2004 [4], which mentioned “Pharmacodependence may develop even at therapeutic doses, and/or for patients who do not show an individualized risk factor”. This measure was not associated with a reduction in problematic use, and the worsening of cases notified was observed at the same time [5]. Along the same lines, a pharmacoepidemiological surveillance study revealed that zolpidem was one of the most present drugs on falsified prescriptions [13], in addition to black market selling. Facing this evidence of problematic consumptions, in 2017, the French national drug regulatory authority stated that zolpidem prescriptions would be subject to part of the regulation of narcotics: prescriptions in full letters on a mandatory secured prescription pad [14]. This new regulation was provided to promote the correct use of zolpidem and limit the risks of abuse and dependence.

Different effects of this regulation were evaluated by the CEIP-A of Nantes, with the ZORRO (ZOlpidem and the Reinforcement of Regulation of prescription Orders) national study, which aimed to evaluate the overall impact of the zolpidem regulatory change on the prescription of hypnotic/sedative and anxiolytic medications [15]. The ZORRO study used an epidemiological approach, combining (i) an analysis of the French national healthcare claims database [16,17] and (ii) a clinical field study, including data collection among general practitioners (GPs) and consumers at risk for zolpidem use. The first results from the healthcare claims database approach showed a decrease of half in zolpidem consumers in the general population, and we identified four specific trajectory patterns in long-term consumers: zolpidem continuation (approximately 40% of the population), zolpidem discontinuation without replacement with another sedative (a third of the population), and zolpidem replacement with zopiclone or another hypnotic benzodiazepine (a quarter of the population) [16,17]. These results, using databases, are valuable for assessing the impact of the regulation on zolpidem delivery in the general population, but they are not sufficient to address other specific questions, such as the impact on specific patterns of zolpidem use by problematic consumers and replacement of zolpidem with nonprescribed medications or licit and illicit psychoactive substances.

Thus, the clinical part of the ZORRO study was expected to be complementary to previous results in evaluating the impact of the new zolpidem regulation on subjects at risk for zolpidem use. It provides an original study design with an individual evaluation of subjects directly recruited at their care centers. Data are scarce in the literature concerning the impact of drug regulations on specific patterns of consumption using clinical data. Thus, the part of the ZORRO study presented here aimed to evaluate the evolution of the problematic consumption of zolpidem at the individual level in two populations at risk of problematic consumption since the new regulation of its prescription.

## 2. Materials and Methods

### 2.1. Study Oversight

This study was a part of the multimodal ZORRO study. The study was funded by a grant from the French National Agency for Medicines and Health Products Safety and was monitored by a pluridisciplinary steering committee with pharmacologists, psychiatrists specialized in addiction, pharmacoepidemiologists, and general practitioners. This study was approved by the French Committee for the Protection of Persons (CPP, 2018-A01070-55) on 11 June 2018 and conducted in accordance with the National Commission of Information Technology and Liberties (CNIL) rules, regarding data management and analysis. The study was registered under the reference NCT03584542, and the study protocol is available as a publication [15].

### 2.2. Study Population

The participants were consumers at risk for zolpidem use who were over 18 years old and had a sufficient level of French fluency to participate. All participants gave their oral consent to participate, in accordance with the Declaration of Helsinki, and recruitment occurred between 8 October 2018 and 28 January 2020.

### 2.3. Study Procedures

Two recruitment channels were used:-General practitioners (GPs) randomly selected nationally or were members of a network of habitual partners of the investigator service identified patients at risk for zolpidem use, according to the substance use disorder (SUD) Diagnostic and Statistical Manual of Mental Disorders 5th edition (DSM-5) criteria [18]. If patients agreed to participate, an anonymous self-questionnaire about zolpidem consumption was given to the patient by the GP. The questionnaire was completed by the patients independently to the GP and placed in a sealed envelope.-Specialized care centers dedicated to drug dependence (SCDDs), located throughout the country, selected users at risk for zolpidem consumption. If the users agreed to participate, they had to complete an anonymous self-questionnaire about their zolpidem consumption.

The same data were collected from the patients recruited by GPs and users recruited by SCDDs. Participants were asked about the following: their sociodemographic data (age, sex); their knowledge about the change in zolpidem prescription rules; evolution in their zolpidem consumption since the regulatory change (continuation without any change, discontinuation, or a reduction with or without replacement); their zolpidem consumption characteristics before the regulatory change (frequency, dose, duration, effects sought and felt, way of obtaining, limitation of the prescription, route of administration); their molecule of replacement and molecule of preference to replace zolpidem, if applicable; their characteristics of consumption of zolpidem or preferred molecule (when replacement) after the regulatory change (same criteria as characteristics of consumption before); and their consumption of other psychoactive substances.

### 2.4. Outcomes

The primary outcome was the evolution of problematic zolpidem consumption since the change in the regulatory framework of zolpidem. For the period before the regulatory change, participants were classified as problematic zolpidem consumers when they met at least one of the following four criteria: overconsumption (>10 mg per day), fraudulent way of obtaining zolpidem, effects sought other than the hypnotic effect, and modes of administration other than oral administration. For the period after the regulatory change, patients were considered as problematic consumers when:-The consumption of zolpidem or another hypnotic in replacement was reported, according to the same four criteria as before the regulatory change.

or

-Replacement with a nonhypnotic substance (medication without hypnotic indication or another psychoactive substance) was reported.

The problematic status of consumers was considered missing when missing data regarding their consumption did not allow for a classification according to our definition.

### 2.5. Statistical Analysis

All statistical analyses were performed for the two populations: patients included by GP offices and users included by SCDDs. The two study populations were first described using counts and percentages for categorical variables and using means and standard deviations for continuous variables. A McNemar test on paired samples was performed, in order to compare the number of problematic consumers before and after the regulatory change. We then described, among patients with a problematic consumption of zolpidem before the regulation, the evolution of the problematic status after the regulatory change, according to the patient’s characteristics and their characteristics of consumption before the regulatory change. Comparisons were performed using Student’s t-test for continuous variables and the chi-squared test or Fisher’s exact test for categorical variables. All analyses were conducted using R software version 3.6.1 (R Core Team, Vienna, Austria) [19].

## 3. Results

Of the 1154 GP offices randomly selected, 113 agreed to include patients at risk for zolpidem use, and 47 included participants. A total of 77 SCDDs were solicited, of which, 59 agreed to participate, and 37 included participants. Eventually, a total of 243 participants were included, 125 by GPs and 118 users from SCDDs.

### 3.1. Description of the Population

A detailed description of the included participants is presented in Table 1, which shows differences between the participants recruited by GP offices and participants recruited from SCDDs.

Missing data < 1% for zolpidem daily dose, zolpidem duration, effects sought with zolpidem and effects felt with zolpidem; missing data from 1 to 2.1% for age, a single administration of zolpidem and mode of obtaining zolpidem; missing data between 4.5 and 8.6% for limitation of prescriptions, alcohol, tobacco, and cannabis consumption; missing data between 14.8 and 19.8% for consumption of other psychoactive substances.

The population from the GP offices was characterized by chronic daily consumption, with nearly half of the patients taking the maximum dosage. For almost all patients, the duration of treatment was superior to those recommended, but recommendations concerning the administration and effects sought and felt were usually respected. It should be noted that almost 20% of the patients had a daily dose superior to the maximum authorized dose, and more than 30% had already experienced limitations of their prescriptions by the GPs. However, very few cases of transgressions to obtain zolpidem were reported.

The population from the SCDDs also corresponded to consumers with daily consumption in the majority of the cases, as well as with frequent chronic use, but with doses usually higher than the authorized maximum. The recommendations for use were not respected in the majority of cases: more than half of the participants used zolpidem outside of bedtime, nearly 20% intended to get high with zolpidem, almost 30% used intravenous administration and some fraudulent manners to obtain zolpidem, in addition to a prescription were reported.

Concerning the consumption of other psychoactive substances, patients from the GP offices had mainly used alcohol and tobacco; the consumption of illicit substances was rare, whereas it was frequently reported by users from SCDDs (50% reported cannabis use and cocaine, and 35% reported heroin use).

Concerning the evolution of zolpidem consumption after the regulatory change, the more prevalent behaviour in the two populations was continuation without any change (62.4% in the GP population and 39.8% in the SCDD population). Two other behaviours were frequently reported: discontinuation or reduction with a replacement (17.6% in the GP population and 28% in the SCDD population) and reduction without any replacement (16% in the GP population and 19.5% in the SCDD population).

In the case of replacement, the experimental substances for zolpidem replacements were mainly medications (95.2% in the GP population and 81.8% in the SCDD population). However, SCDD users also reported the use of other licit types of illicit substances for the replacement of zolpidem (21.2% reported cannabis use, 18.2% reported alcohol use, 15.2% reported heroin use, and 12.1% reported tobacco use). Regarding the preferred drug for the replacement of zolpidem, medications with a hypnotic indication were frequently reported in both populations (more than one-third of the users), with zopiclone being the more frequently cited molecule. Medications with an anxiolytic indication were the second-most preferred substances.

### 3.2. Primary Outcome: Evolution of Problematic Zolpidem Consumption after the Regulatory Change

The prevalence and criteria for problematic use before and after the regulatory change for zolpidem prescriptions are presented in Table 2. Before the regulatory change, the prevalence of problematic use was much higher in users recruited from SCDDs than in patients recruited from GP offices. Both populations had the same most reported criterion: overconsumption (more than 3/4 of the problematic users). The second-most reported criterion by problematic users was the fraudulent way of obtaining zolpidem (one-third among GP office users and three-quarters among SCDDs users). After the regulatory change, the prevalence of problematic consumption (zolpidem or a replacement substance) decreased from 24.8% to 20.8% for patients from GP offices (McNemar’s test *p* = 0.593, Table 3 and from 73.7% to 51.7% for users recruited from SCDDs (McNemar’s test *p* < 0.001, Table 4). Regarding the criteria for problematic consumption after the regulatory change, the prevalence of each criterion was close to the prevalence before the regulatory change, except for the prevalence of fraudulent ways of obtaining zolpidem, which decreased for both populations. For both populations, the majority of the participants had the same status but in an inverted proportion: 65.6% of the patients from GP offices continued to have no problematic use, and 50.8% of the users from SCDDs continued to have problematic use. Evolution from no problematic use to problematic use was rare and concerning in the majority of users who preferentially replaced zolpidem with a nonhypnotic substance.

### 3.3. Characteristics of the Participants who Remained Problematic Consumers after the Regulatory Change

The sociodemographic characteristics, consumption of other psychoactive substances, and characteristics of zolpidem use before the regulatory change were compared between users who were still categorized as problematic users after the measure and those who were not.

In SCDD users, a longer duration of zolpidem use before the regulatory change was associated with problematic users after the change (*p* = 0.013). Users of other psychoactive substances were also less likely to continue to have problematic use after the regulatory change (Table 5).

In GP users, we found that patients with a fraudulent way of obtaining zolpidem before the regulatory change were less likely to be categorized as problematic users after the change (*p* = 0.022). Patients with a long duration of zolpidem use before the regulatory change seemed to remain problematic users after the change (*p* = 0.123) (data not shown).

## 4. Discussion

### 4.1. Main Results

This clinical part of the ZORRO study assessed the evolution of the problematic consumption of zolpidem after the new regulation for zolpidem prescriptions from a patient-centred evaluation among at-risk consumers in two distinct populations. Evolution of the prevalence of problematic consumers since the new regulation showed mitigated results: it barely decreased (a quarter to a fifth) in the GP-recruited patients, whereas it fell from three-quarters to 52% in SCDD users. Some characteristics were found to be associated with the status of remaining problematic, such as a long duration of the treatment before the change or, in SCDD users, a trend for less frequent consumption of other psychoactive substances.

### 4.2. Characteristics of Two Different Populations

The differences found in the evolution of problematic use in the two populations need to be interpreted in light of the differences in consumer profiles. The population of GP patients at risk for zolpidem use was mostly represented by older women who sought the hypnotic effect of zolpidem with a “proper” use (bedtime by the oral route), but with chronic consumption, by far exceeding the regulatory month of prescription. The same difficulty of withdrawing zolpidem (almost two-thirds continued zolpidem use without any change) that this study highlights was also noticed in a previous study with a sample of elderly patients who shared the characteristics of the sex ratio and duration of prescription [20,21]. It is well-known that the long duration of prescription is a major risk factor for failing to stop using benzodiazepines and related drugs, such as zolpidem [21,22,23]. Nevertheless, this difficulty more pronounced for the GP population can appear slightly paradoxical, given that the subjects were more informed by their GPs, concerning the new regulation, and were less frequently categorized as problematic consumers before the regulation.

The second population represented younger males who, as a majority, also sought the hypnotic effect of zolpidem, but also sought anxiolytic effects or, likewise, the “get high” effects. Consumption in this group differs from the proper use with a majority of higher dosage intake than the daily recommended dose and intake that can occur throughout the whole day. Although they were recruited in structures that traditionally manage consumers with more severe substance use disorders, they were more prone than the first population to stop or reduce zolpidem use. A major explanatory factor is that SCDD patients were probably consulted from the perspective of receiving specialized addictive care, whereas recruitment by GPs was performed for all patients, regardless of the purpose of the consult. Another factor to consider: SCDD users frequently reported other psychoactive substance consumption. We can hypothesize that it was less difficult for SCDD users to stop or reduce zolpidem consumption with an available and already known substance (maybe taken more frequently and/or in higher amounts) that could “substitute” the effects sought with zolpidem. Indeed, we found that patients using other psychoactive substances were less likely to remain problematic users after the change.

### 4.3. Reinforcement of Prescriptions and Fraudulent Ways of Obtaining Zolpidem

The fraudulent ways of obtaining were found to be a major criterion of problematic use. However, we found that for patients who pursued zolpidem consumption, there was a decrease in the fraudulent ways of obtaining zolpidem. This was more marked in the SCDD population but was also effective in the GP recruitment population. Among the fraudulent ways of obtaining, prescription falsification was a potential target of the mandatory secured prescription pad. Thus, the decrease observed here for falsified prescriptions is coherent with the *Ordonnances Suspectes Indicateur d’Abus Possible* (OSIAP) study results, where zolpidem fell from the first place in 2015–2017 to the fourth place in 2018 [24]. However, a falsified prescription was not the main fraudulent way of obtaining zolpidem in our study (donation, black market, doctor shopping, etc.). Thus, we can say that this new regulation has reduced zolpidem prescriptions (seen in the database study [16]) and reduced the quantity of zolpidem circulating in fraudulent ways (this study). The other criteria were less impacted by the new regulation.

### 4.4. Strengths and Weaknesses

The unique multimodal design of the ZORRO study combining complementary approaches (a clinical approach and a database approach) provides valuable knowledge regarding the overall impact of the health regulation. No clear guidelines are provided in the methodology application to assess the impact of a regulatory action at an individual level. Indeed, two systematic reviews in 2013 and 2018 investigated the scientific literature regarding methodologies used to assess the impact of regulatory actions, notably for the Food and Drug Administration in the USA and Europe [25,26], and revealed the heterogeneity of the methodologies employed. None of the studies focused on hypnotics/zolpidem, and only a few studies concerned a patient-centred evaluation. In these studies, the data collected were not compared to medico-administrative data and, to our knowledge, never compared from a global perspective integrating database and individual evaluations. This clinical part of the ZORRO study is the first to assess the impact of a health regulation on specific populations of problematic zolpidem consumers. This field study provides complementary results to previous results on a database regarding (i) a specific population of zolpidem users at risk and (ii) clinical data about consumption that are unavailable in medico-administrative data (type of consumption and use of nonprescribed substances). We especially identified that some licit or illicit substances for replacing zolpidem were reported by users in SCDDs. However, the majority of the time, the preferred substance was a medication in GP offices and SCDDs. This explains why we found a close proportion of patients with replacement substances in a context of discontinuation or reduction of zolpidem in this clinical study, compared to our previous results using a healthcare database (18% in GP offices and 28% in SCDDs versus a quarter in database study [17]). However, we found a higher proportion of patients who continued zolpidem use (any change or reduction without a replacement), compared to the healthcare database results (80% in GP offices and 64% in SCDDs versus approximately 40%). The main factor appears to be the differences between the study populations: at-risk users of problematic zolpidem consumption in our study versus more broadly chronic users in the healthcare database. We can assume that at-risk zolpidem users may have much difficulty stopping zolpidem and, moreover, stopping zolpidem without a drug replacement. Taking this into account, off-prescription zolpidem intake in our study may also have played a role in the difference, but to a lesser extent because it was only reported by SCDD users.

A major limitation is the relatively small size of the study populations, which can limit the generalizability of our results. However, the profiles of problematic zolpidem consumers found in GP offices and SCDDs were similar to those described in previous reports [4,5,10]. In the two channels of recruitment, the inclusion criteria were at-risk users of zolpidem. However, we can suspect that risky consumption may have been differently interpreted by GPs and SCDDs, given the heterogeneity between the two populations. This may have slightly contributed to the difference in the prevalence of problematic consumption among at-risk users. Declarative bias was also possible, especially regarding fraudulent ways of obtaining zolpidem or the underreporting of illicit consumption. The use of anonymous self-questionnaires in a sealed envelope may have limited underreporting. Moreover, the evaluation of subjects, which occurred from 1.5 years to 3 years after the regulation, may have led to a memory bias.

## 5. Clinical Implications

In our study, we individualize different results, in terms of impact on problematic zolpidem consumption, according to patient characteristics and prior consumption typology. Thus, the clinical implications for practitioners caring for patients with problematic zolpidem consumption will be the integration of the clinical characteristics of the patient (other substances abuse and dependence), as well as the type of consumption (duration, dosage, route of administration, sought effect, etc.), in order to individually optimize the therapeutic management and ensure the correct application of the regulation. Indeed, the new regulation of zolpidem had secure prescription as a lever of action and applied to all patients, regardless of their clinical characteristics or consumption typology. However, adherence by prescribers to a regulation or a recommendation may be limited by patient-related factors, such as the non-integration of the unique characteristics of patients in these recommendations [27]. In a study of the ZORRO project on the perception of this new regulation by GPs, we showed that patient-related parameters (duration of zolpidem intake, presence of psychiatric comorbidity, age, etc.) were associated with the choice to continue or not the prescription of zolpidem [28]. This was concordant with a study with French health insurance data where the decrease in zolpidem reimbursement was more pronounced in younger than older subjects [29].

## 6. Conclusions

This clinical study between two different populations of problematic consumers revealed that the new French regulation of zolpidem prescriptions had an impact on the fraudulent ways of obtaining zolpidem. In order to mitigate the results of the evolution of problematic use on the GPs population, we must strengthen the vigilance of clinicians on zolpidem dependence, and the results in the SCDD population warn us of the necessity of helping patients quit zolpidem use, while avoiding switching to an alternative addiction.

## Figures and Tables

**Table 1 ijerph-19-08920-t001:** Characteristics of the participants included in the study.

Variables	Patients from GP Offices *n* = 125 Number (%) or Mean (sd)	Users from SCDDs *n* = 118 Number (%) or Mean (sd)
**Sociodemographic data**		
Age (years)	65.9 (13.8)	41.8 (11.9)
Sex (men)	38 (30.4)	86 (72.9)
**Knowledge about the regulatory change for zolpidem (yes)**	101 (80.8)	89 (75.4)
By whom? *	*n* = 101	*n* = 89
Practitioner	93 (92.1)	68 (76.4)
Pharmacist	17 (16.8)	25 (28.1)
Other	1 (1.0)	9 (10.1)
**Zolpidem consumption before the regulatory change**		
Daily consumption (yes)	110 (88.0)	99 (83.9)
Daily dose		
Number of tablets	1.1 (0.5)	5.3 (9.2)
10 mg	72 (58.1)	35 (29.9)
>10 mg	24 (19.4)	79 (67.5)
Single administration (yes)	101 (83.5)	73 (62.4)
Only taken at bedtime (yes)	104 (83.2)	55 (46.6)
Duration		
<1 month	1 (0.8)	1 (0.9)
1 to 3 months	2 (1.6)	16 (13.7)
4 months to 3 years	25 (20.2)	42 (35.9)
4 to 10 years	41 (33.1)	36 (30.8)
More than 10 years	55 (44.4)	22 (18.8)
Effect sought *		
Sleeping	123 (98.4)	97 (82.9)
Anxiolysis, calming	6 (4.8)	21 (17.9)
Get high	0 (0.0)	23 (19.7)
Other	5 (4.0)	3 (2.6)
Effects felt *		
Sleeping	120 (96.8)	90 (76.3)
Anxiolysis, calming	6 (4.8)	28 (23.7)
Get high	1 (0.8)	20 (16.9)
Other	2 (1.6)	14 (11.9)
None	2 (1.6)	2 (1.7)
Route of administration *		
Per os	125 (100.0)	95 (80.5)
IV	0 (0.0)	32 (27.1)
Nasal	0 (0.0)	7 (5.9)
Method of obtaining *		
Prescription	125 (100)	113 (95.8)
Black market	2 (1.6)	20 (17.2)
Falsification of a prescription	0 (0.0)	18 (15.5)
Medical nomadism	4 (3.2)	24 (20.7)
Donation	4 (3.2)	26 (22.4)
Other	0	4 (3.4)
Limitation of prescriptions by GPs (yes)	41 (34.7)	42 (38.5)
**Consumption of other psychoactive substances (at least once in the past year) ***		
Alcohol	61 (52.1)	81 (71.1)
Tobacco	23 (20.0)	101 (86.3)
Cannabis	5 (4.5)	59 (52.7)
Cocaine	0 (0.0)	56 (50.0)
Heroin	1 (1.0)	38 (34.9)
Amphetamines	0 (0.0)	19 (18.3)
Hallucinogens	0 (0.0)	21 (20.2)
NPS	0 (0.0)	12 (12.0)
Others	1 (1.1)	24 (30.8)
**Evolution of zolpidem consumption after the regulatory change**		
Continuation without any change	78 (62.4)	47 (39.8)
Discontinuation without any replacement	5 (4.0)	10 (8.5)
Reduction without any replacement	20 (16.0)	23 (19.5)
Discontinuation or reduction with a replacement	22 (17.6)	33 (28.0)
Increase of zolpidem consumption	0 (0.0)	5 (4.2)

%, percentage; sd, standard deviation; GP: general practice; SCDD: specialized care centre dedicated to drug dependence; NPS: new psychoactive substances; IV: intravenous. * Several possible answers for one participant.

**Table 2 ijerph-19-08920-t002:** Prevalence and criteria for problematic consumption before and after the regulatory change.

	Patients from GP Offices *n* = 125 Number (%)	Users from SCDDs *n* = 118 Number (%)
Problematic consumption of zolpidem BEFORE the regulatory change		
Yes	31 (24.8)	87 (73.7)
Overconsumption	24 (77.4)	79 (91.9)
Fraudulent way of obtaining zolpidem	9 (29.0)	64 (74.4)
Effects other than hypnotic effects sought	7 (22.6)	22 (25.3)
Modes of administration other than oral administration	0 (0.0)	37 (42.5)
No	93 (74.4)	30 (25.4)
Missing data	1 (0.8)	1 (0.8)
Problematic consumption of zolpidem AFTER the regulatory change		
Yes	26 (20.8)	61 (51.7)
Continuation of zolpidem or replacement by another hypnotic	19	47
Overconsumption	14 (73.7)	42 (89.4)
Fraudulent way of obtaining zolpidem	2 (10.5)	25 (58.1)
Effects sought other than hypnotic effects	5 (26.3)	14 (32.6)
Modes of administration other than oral administration	0 (0.0)	16 (37.2)
Replacement by a nonhypnotic substance	7	14
No	90 (72.0)	46 (39.0)
Missing	9 (7.2)	11 (9.3)

%, percentage. GP: general practice; SCDDs: specialized care centers dedicated to drug dependence. Missing data < 1% overconsumption and fraudulent ways of obtaining zolpidem before the regulatory change; missing data equal to 6.1 % for fraudulent ways of obtaining zolpidem effects, other than hypnotic effects sought and modes of administration other than oral administration.

**Table 3 ijerph-19-08920-t003:** Comparison of problematic use before and after the regulatory change for GP office patients’ problematic users (*n* = 125).

		AFTER			
		**Yes**	**No**	**Missing**	**Total**
BEFORE	Yes	20 (16.0%)	8 (6.4%)	3 (2.4%)	31 (24.8%)
	No	6 (4.8%)	82 (65.6%)	5 (4.0%)	93 (74.4%)
	Missing	0	0	1 (0.8%)	1 (0.8%)
	Total	26 (20.8%)	90 (72.0%)	9 (7.2%)	125 (100.0%)

McNemar’s test = 0.593.

**Table 4 ijerph-19-08920-t004:** Comparison of problematic use before and after the regulatory change for SCDD problematic users (*n*= 118).

		AFTER			
		**Yes**	**No**	**Missing**	**Total**
BEFORE	Yes	60 (50.8%)	18 (15.3%)	9 (7.6%)	87 (73.7%)
	No	1 (0.8%)	27 (22.9%)	2 (1.7%)	30 (25.4%)
	Missing	0	1 (0.8%)	0	1 (0.8%)
	Total	61 (51.7%)	46 (39.0)	11 (9.3%)	118 (100.0%)

McNemar’s test < 0.001.

**Table 5 ijerph-19-08920-t005:** Comparison of problematic and nonproblematic users after the regulatory change among SCDD users who had problematic consumption of zolpidem before the regulatory change (*n* = 87).

	Non-Problematic Users *n* = 18 Number (%) or Mean (sd)	Problematic Users *n* = 60 Number (%) or Mean (sd)	*p*-Value
Age (years)	40.5 (10.8)	40.7 (11.0)	0.953
Sex (men)	16 (88.9)	40 (66.7)	0.124
Duration of zolpidem consumption before the regulatory change			
3 months or less	6 (33.3)	6 (10.0)	0.013 *
4 months to 3 years	9 (50.0)	20 (33.3)	
4 to 10 years	2 (11.1)	20 (33.3)	
More than 10 years	1 (5.6)	14 (23.3)	
Consumption of other psychoactive substances (at least once in the past year)			
Alcohol	14 (82.4)	37 (63.8)	0.251
Tobacco	18 (100.0)	48 (81.4)	0.111
Cannabis	13 (76.5)	31 (55.4)	0.202
Cocaine	12 (70.6)	27 (47.4)	0.160
Heroin	7 (41.2)	18 (33.3)	0.765
Amphetamines	5 (29.4)	8 (16.0)	0.289 *
Hallucinogens	7 (41.2)	9 (17.6)	0.095 *
NPS	2 (11.8)	5 (10.9)	1 *
Criteria for problematic consumption before the regulatory change			
Overconsumption	16 (88.9)	55 (93.2)	0.922
Fraudulent ways of obtaining zolpidem	12 (70.6)	43 (71.7)	1
Effects sought other than hypnotic effects	7 (38.9)	14 (23.3)	0.316
Modes of administration other than oral administration	8 (44.4)	25 (41.7)	1

%, percentage; sd, standard deviation; NPS: new psychoactive substances; * Fisher’s exact test; missing data = 1.1% for age, tobacco consumption, overconsumption, and fraudulent ways of obtaining zolpidem, 3.4% for alcohol consumption, 4.6% for cocaine consumption, 5.7% for cannabis consumption, 9.2% for heroin consumption, 12.6% for hallucinogen consumption, 13.8% for amphetamine consumption, and 18.4% for NPS consumption.

## Data Availability

Given the confidentiality of the data, our ethics committee (GNEDS) is preventing us from making our data set publicly available. However, we are willing to make our data available upon request as we consider that it is important for open and reproducible science, and thus we will ensure that all interested and qualified researchers will be able to be granted access. Furthermore, vosdonneespersonnelles@chu-nantes.fr is the contact for data requests to approve and distribute our data.

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
