# Peer review of "Did the New French Regulation of Zolpidem Decrease the Problematic Consumption of Zolpidem? A Field Study among Users"

_ijerph, 2022, doi:10.3390/ijerph19158920_

Round 1

Reviewer 1 Report

In my opinion, the introduction to the study lacked basic information on the mechanism of action of zolpidem, as well as explaining the risks associated with long-term use of this drug.

Zolpidem is a hypnotic medicine from the imidazopyridine group. Unlike benzodiazepines that bind indiscriminately to all three types of benzodiazepine receptors (omega-1, -2 and -3), zolpidem binds only to the omega-1 receptor - it is a selective agonist of the benzodiazepine type 1 receptor. The omega-1 receptor is part of a receptor called GABA-A, which has the ability to bind many substances, including GABA. Zolpidem acts indirectly by increasing the ability of GABA to bind to the GABA-A receptor. Due to the inhibitory effect of GABA on nerve cells, the drug has an indirect inhibitory effect on the central nervous system.

Zolpidem shows a rapid hypnotic effect (it helps to fall asleep, extends the total duration of sleep, prevents the phenomenon of early awakening and reduces the number and duration of night awakenings). Compared to benzodiazepines, it provides deep sleep phases (phases 3 and 4 of slow wave sleep).

Zolpidem is indicated for the short-term (up to 4 weeks) treatment of insomnia.

It should be clearly emphasized in the introduction to the manuscript that zolpidem has a high addictive potential.

Administering the drug for several weeks leads to the development of tolerance and thus reduces its effectiveness. Long-term use of the drug may cause physical and psychological dependence. Of course, the risk of dependence increases with dose and duration of therapy. It is greater in people who have been addicted to alcohol or medication, and in people with personality disorders. Sudden withdrawal may trigger the so-called withdrawal symptoms (e.g. anxiety, agitation, dizziness, headache, muscle aches, tension, irritability).

In general, the prescription of addictive hypnotic drugs is abused all over the world. Therefore, it is worth publishing information monitoring this phenomenon.

Minor remarks:

Verse 116: between "recruited" and "by" - more than one space

Verse 139: no full stop at the end of the sentence

Author Response

We thank the reviewer for his comment on our article. In accordance with his recommendations, we have added in the first paragraph of the introduction information on the mechanism of action of zolpidem (l. 37 – 42) as well as on its dependence potential (l. 32 – 36) in addition to what was already mentioned.

We also corrected the typos mentioned by the reviewer as well as others along the text.

We fully agree with the reviewer on the importance of the phenomena of abuse and dependence concerning zolpidem. This molecule is effectively monitored at the national level in France and has given rise to several regulations from the health authorities. This underlines the importance of a large multimodal study as the ZORRO project that is monitoring the consequences on consumption after the new regulation with epidemiological and clinical outcomes. This project was funded by the ANSM (the French National Agency for Medicines and Health Products Safety) and has been registered as a clinical research project on clinicaltrials (NCT03584542).

Reviewer 2 Report

The article submitted by Laforgue et al., entitled by “Did the new French regulation of zolpidem decrease the problematic consumption of zolpidem? A field study among users” (ijerph-1764553), investigated two different populations of problematic consumers to explore the effect of new French regulation of zolpidem prescriptions. This article is basically not an academic paper but an investigation report. The reason and mechanism are rarely disclosed. In this reviewer’s opinion, this article is not suitable for the publication in the academic journal, Int. J. Environ. Res. Public Health. 

Author Response

We consider our work as original research. Indeed, it highlights differences in patient characteristics that have clinical implications in the management of patients with problematic zolpidem use. Indeed, we summarized in the Discussion part the findings and the reasons and mechanisms that can explain point by point our results:

- The differences in subject characteristics in the two study populations

- The differences in the profile of zolpidem problematic use in the two study populations

- The impact on the fraudulent way of obtaining (in relation to the type of regulatory change: secure prescription pads implementation).

The presence of patient-related factors is often cited in the literature by physicians as a barrier to the proper use of drugs and the correct application of recommendations. Thus, the highlighting of different results according to the characteristics of the patients in our work underlines the need for the doctors taking charge of these patients of a preliminary evaluation assessing these parameters. To clarify this point, we have added a paragraph "clinical implications" in the manuscript (l. 406 – 421). Also, as suggested by reviewer 1, we added data on zolpidem mechanism of action and dependence potential in the introduction.

This clinical study is part of the global project ZORRO using a multimodal approach to assess the impact of the regulatory change regarding to zolpidem (clinical data and health insurance database), as detailed in the Clinicaltrials.gov. Each study of this project aimed to answer to specific research questions regarding to the impact of regulatory change. The study presented here addresses the question of the outcome of problematic consumers after the new regulation with real-life patients-related outcomes and is thus a clinical research study. However, these research questions are complementary to interpret the impact on a global point of view and that is the big strength of the project ZORRO. That is why we also compared and contextualized our results regarding to the others already published in the context of the project.

The question of the impact of regulations on patients is of major public health interest as has been discussed by certain papers cited in references in the manuscript. In our view, our article with its real-life data thus falls within the scope of the IJERPH journal and particularly of the special issue "New Perspectives in Real-World Pharmacoepidemiology and Drug Safety" with the objective of "to increase awareness among health decision makers concerning the epidemiological and clinical impact of an inappropriate prescription".

Reviewer 3 Report

Dear authors,

the manuscript is very interesting and well-edited in English. Due to its prevalence, this problem is often overlooked. We need to work on the bias toward the changes we want to introduce. This article fulfills that role. The new French regulations have an impact on the taking of zolpidem. However, they cannot eliminate overconsumption of this medication. 

Author Response

We thank the reviewer for his appreciation of our work.

Round 2

Reviewer 2 Report

The revised is better than before, and I recommend its publication. 

This manuscript is a resubmission of an earlier submission. The following is a list of the peer review reports and author responses from that submission.